# Context Shift Reduction for Offline Meta-Reinforcement Learning

**Yunkai Gao[1,2,3]**   **Rui Zhang[2]**   **Jiaming Guo[2]**   **Fan Wu[2,3,4,5]**   **Qi Yi[1,2,3]**
**Shaohui Peng[5]**   **Siming Lan[1,2,3]**   **Ruizhi Chen[5]**   **Zidong Du[2,6]**   **Xing Hu[2,6]**
**Qi Guo[2]**   **Ling Li[4,5]**   **Yunji Chen[2,4]** *

[1] University of Science and Technology of China, USTC, Hefei, China
[2] State Key Lab of Processors, Institute of Computing Technology,
Chinese Academy of Sciences, Beijing, China
[3] Cambricon Technologies
[4] University of Chinese Academy of Sciences, UCAS, Beijing, China
[5] Intelligent Software Research Center, Institute of Software, CAS, Beijing, China
[6] Shanghai Innovation Center for Processor Technologies, SHIC, Shanghai, China
{gyk314, yiqi, lansm}@mail.ustc.edu.cn
{wufan2020, pengshaohui, ruizhi, liling}@iscas.ac.cn
{zhangrui, guojiaming, duzidong, huxing, guoqi, cyj}@ict.ac.cn

## Abstract

Offline meta-reinforcement learning (OMRL) utilizes pre-collected offline datasets to enhance the agent's generalization ability on unseen tasks. However, the context shift problem arises due to the distribution discrepancy between the contexts used for training (from the behavior policy) and testing (from the exploration policy). The context shift problem leads to incorrect task inference and further deteriorates the generalization ability of the meta-policy. Existing OMRL methods either overlook this problem or attempt to mitigate it with additional information. In this paper, we propose a novel approach called **C**ontext **S**hift **R**eduction for **O**MRL (**CSRO**) to address the context shift problem with only offline datasets. The key insight of CSRO is to minimize the influence of policy in context during both the meta-training and meta-test phases. During meta-training, we design a max-min mutual information representation learning mechanism to diminish the impact of the behavior policy on task representation. In the meta-test phase, we introduce the non-prior context collection strategy to reduce the effect of the exploration policy. Experimental results demonstrate that CSRO significantly reduces the context shift and improves the generalization ability, surpassing previous methods across various challenging domains.

## 1   Introduction

Meta-Reinforcement Learning (RL) has made significant strides in enhancing agents' generalization capabilities for unseen tasks. Meta-RL methods [6, 5, 27] leverage vast amounts of trajectories gathered across a task set during the meta-training phase to learn the shared structures of tasks. During meta-testing, these meta-RL methods exhibit swift adaptation to new tasks with a limited number of trials. Context-based meta-RL methods [27, 39, 7] have emerged as the most popular approaches, showcasing remarkable performance and rapid task adaptation. These methods first acquire task information by learning a task representation from the context (i.e., the set of transitions) and subsequently condition the policy on the task representation to execute appropriate actions.

---

*Corresponding Author.

37th Conference on Neural Information Processing Systems (NeurIPS 2023).

Due to the requirement of a vast number of trajectories sampled from a task set, data collection in meta-RL can be both expensive and time-consuming. Offline RL [17, 32, 14] presents a potential solution to mitigate the cost of data collection. These methods only utilize large datasets that were previously collected by some policies (referred to as the behavior policy), without any online interaction with the environment during the training phase.

By combining meta-RL and offline RL, offline meta-reinforcement learning (OMRL) [18, 20, 37] amalgamates the benefits of generalization ability on unseen tasks and training without requiring interaction with the environment. During meta-training, both the acquisition of task representation and the training of the policy utilize pre-collected offline trajectories obtained from the task-dependent behavior policy.

However, OMRL faces a context shift problem: a distribution discrepancy arises between the context obtained from the behavior policy during meta-training and the context collected from the exploration policy during meta-testing [26]. Given that the task representation is learned from the context, this context shift problem hinders the accurate inference of task information by the task representation. Consequently, this leads to a decline in performance when adapting to unseen tasks. As shown in Table 1, we compared the test results of FOCAL [20] and OffPearl [27] in two different environments and found that context shift significantly reduced testing performance.

Existing OMRL methods primarily aim to incorporate the benefits of offline RL into meta-RL, but they do not fully address the context shift problem. Some approaches [20, 21, 19] utilize pre-collected offline datasets from the test tasks to acquire task representation, disregarding the context shift problem. However, this approach is impractical as obtaining pre-collected offline datasets for new and unseen tasks is extremely challenging. Other methods attempt to mitigate the context shift problem but require additional information beyond the use of offline datasets alone. For instance, [4] assumes that the reward functions of all tasks are known, [26] requires the collection of a substantial volume of online data post offline training.

In this paper, we introduce a novel approach called **C**ontext **S**hift **R**eduction for **O**MRL (**CSRO**) to address the context shift problem using only offline datasets. The key insight of CSRO is to minimize the impact of the policy in context during both the meta-training and meta-test phases. Firstly, the context is collected from the task-dependent behavior policy, as depicted in Figure 1 left. When encoding the context into the task representation during meta-training, the task representation tends to embed the characteristics of the behavior policy. This incorrect task inference during the meta-test phase stems from the disparity between the behavior policy and the exploration policy. Hence, our objective is to reduce the influence of the behavior policy on task representation during the meta-training phase. To achieve this, we design a max-min mutual information representation learning mechanism that minimizes the mutual information between the task representation and the behavior policy, while maximizing the mutual information between the task representation and the task information. During the meta-test, the current exploration policy of OMRL [27, 4, 26] conditions on an initial task representation. The collected context corresponds to the policy of the initial task representation, which significantly differs from the behavior policy, as shown in Figure 1 middle. This context leads to incorrect task inference and control, as depicted in Figure 1 right. Therefore, we propose the non-prior context collection strategy to reduce the impact of the exploration strategy. In practice, the agent initially explores the task independently and randomly at each step, then the agent proceeds to explore based on the context. Through random exploration, the agent initially gains a preliminary understanding of the task and gradually enhances the accuracy of task recognition during subsequent exploratory steps.

Experimentally, we compare the proposed CSRO with previous OMRL algorithms. For a fair comparison, we use the same offline backbone algorithm and the same offline data for environments including Point-Robot and MuJoCo physics simulator [29] with reward or dynamic function changes. Experimental results demonstrate that the proposed CSRO can effectively mitigate the context shift problem and enhance the generalization capabilities, outperforming prior methods on a range of challenging domains.

## 2 Related Works

**Meta Reinforcement Learning.** Meta-RL leverages training task sets sampled from the task distribution to learn and generalize to new tasks. Gradient-based meta-RL [6, 33] methods seek

Table 1: Context shift leads to testing performance decline. Context A refers to the use of offline data collected by behavior policy as the context during testing, with no context shift problem. Context B, on the other hand, involves using a trained meta-policy to explore and collect context based on an initial task representation, resulting in a context shift problem.

| Env | Point-Robot | | Half-Cheetah-Vel | |
|---|---|---|---|---|
| | context A | context B | context A | context B |
| FOCAL | **-4.4**±0.1 | -14.9±1.1 | **-45.7**±2.7 | -69.5±9.6 |
| OffPearl | **-5.1**±0.1 | -17.8±1.5 | **-123.0**±11.5 | -162.8±28.8 |

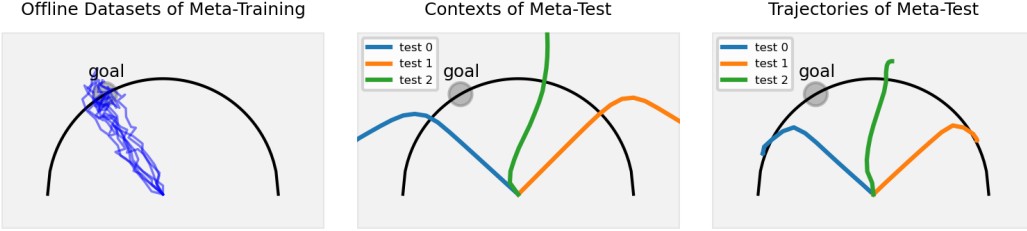

Figure 1: Offline meta-RL on the Point-Robot: one of the tasks where the task objective is to control the agent to reach the goal on the semicircle. **Left:** During meta-training, the offline dataset of this task is collected by task-dependent behavior policy and comprised solely of trajectories leading towards the goal. **Middle:** The agent was trained using the OffPearl [27] method. During the meta-test phase, the agent utilizes the current exploration policy of OMRL [27, 4, 26] and employs the learned meta-policy, which is conditioned on an initial task representation sampled from a prior distribution, to collect contexts. **Right:** The agent navigates out trajectory based on the context depicted in the middle subfigure, the same color represents the corresponding context and trajectory.

policy parameter initialization that requires only a few update steps to generalize well on the new tasks. Since the parameters that cause reward or dynamic changes across different tasks are unknown, context-based meta-RL [27, 39, 16, 11] address this problem as a PODMP. These methods employ a context encoder to extract task representations from historical trajectories, enabling the identification of task domains, with policies conditioned on these acquired task representations. Our proposed CSRO is based on context-based meta-RL frameworks [27].

**Offline Reinforcement learning.** Unlike on-policy and off-policy RL, in offline RL, the agent does not interact with the environment during the training process and only interacts with the environment during the test phase. Offline RL methods [15, 14, 12, 8] use pre-collected data generated by behavior policy for training, effectively reducing the cost of data collection. the distribution shift between the behavior policy and the learned policy introduces the challenge of Q overestimation [17]. [32, 8, 13] make the distribution shift bounded by constraining the differences between learned policy and behavior policy. We employ BRAC [32] as our offline backbone algorithm.

**Offline Meta Reinforcement learning.** OMRL solves the generalization problem and expensive data collection problem by using meta-learning on pre-collected offline datasets. Existing OMRL methods [20, 18, 4, 37, 31] use historical trajectories from offline datasets to infer task representations during meta-training. In the meta-test phase, existing methods can be categorized into two types based on the source of the context: offline test and online test. Offline test methods [18, 20] use sampled context from the pre-collected offline data in the test task, which is impractical and ignores the context shift problem. In contrast, online test methods [4, 26] use the context collected by the agent in the test task. While these online test methods can mitigate the context shift problem, they require additional information beyond just offline datasets, such as reward functions of all tasks [4] and additional online data [26]. We focus on improving the generalization performance of the online test by addressing the context shift problem using fully offline datasets.

**Mutual Information Optimization.** In order to learn more about essential representation, mutual information optimization is often employed to enhance the correlation between the representation

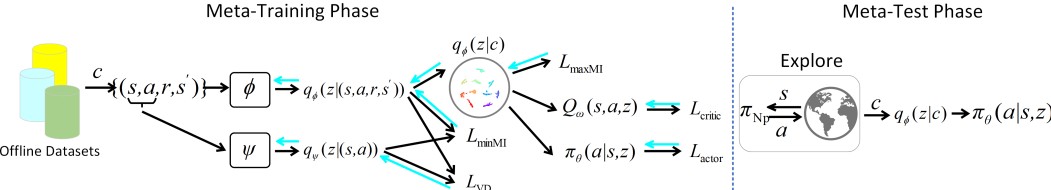

Figure 2: CSRO framework. **Left:** In meta-training, the context encoder $q_\phi$ extracts each transition embedding from $(s, a, r, s')$ transition tuple sample from offline datasets, the CLUB variational distribution estimator $q_\psi$ is used to approximate $p(z|(s, a))$. We use the mean of transition embedding as task representation which conditions both the critic $Q_\omega$ and the actor $\pi_\theta$. **Right:** In test phase, agent uses $\pi_{Np}$ to explore new tasks.

and other properties. The greater the mutual information, the greater the correlation between the two variables. [2, 22] maximizes the mutual information between variables by maximizing the estimated mutual information lower bound. [1, 3] minimizes the mutual information between variables by minimizing the upper bound of the estimated mutual information. In this work, we utilize CLUB [3] to minimize the mutual information between task representation and behavior policy.

## 3 Preliminaries

**Problem Formulation.** In RL, we formalize each task as a Markov Decision Process (MDP), defined by $M = (S, A, P, \rho_0, R, \gamma)$ with state space $S$, action space $A$, transition function $P(s'|s, a)$, reward function $R(s, a)$, initial state distribution $\rho_0(s)$ and discount factor $\gamma \in (0, 1)$. The distribution of policy is $\pi(a|s)$. We aim to maximize the expected cumulative rewards $J(\pi) = E_\tau[\sum_{t=0}^\infty \gamma^t R(s_t, a_t)]$, where $\tau$ is trajectory following $s_0 \sim \rho_0$, $a_t \sim \pi(a_t|s_t)$, $s_{t+1} \sim P(s_{t+1}|s_t, a_t)$. We define the value function and action-value function as:

$$V_\pi(s_t) = \mathbb{E}_{a_t \sim \pi, s_{t+1} \sim P(s_{t+1}|s_t, a_t)}[\sum_{l=0}^\infty \gamma^l R(s_{t+l}, a_{t+l})] \tag{1}$$

$$Q_\pi(s_t, a_t) = R(s_t, a_t) + \gamma \mathbb{E}_{s_{t+1} \sim P(s_{t+1}|s_t, a_t)}[V_\pi(s_{t+1})] \tag{2}$$

In offline meta-reinforcement learning (OMRL), we assume each task sampled from a prior task distribution, $M_i = (S, A, P_i, \rho_0, R_i, \gamma) \sim p(M)$. Different tasks have the same state space and action space, but the reward function or transition dynamic is different due to different parameters, So the task distribution can be defined as $p(R, P)$. The offline datasets $D_i = \{(s_{i,j}, a_{i,j}, r_{i,j}, s'_{i,j})\}_{j=1}^{N_{size}}$ of task $M_i$ is composed of the state, action, reward, and next-state tuples collected by the task-dependent behavior policy $\pi_\beta^i(a|s)$. Since the parameters of different tasks are unknown, for each task, context-based OMRL samples a mini-batch of historical data from the offline datasets as the context $c = \{(s_j, a_j, r_j, s'_j)\}_{j=1}^n$. The context encoder transforms the context $c$ into a task representation $z = q_\phi(z|c)$. Policy, value function, and action-value function are all conditioned on $z$. We use offline data to train the meta policy $\pi_\theta(a|s, z)$ with parameters $\theta$, and will not interact with the environment during the training. OMRL aims to maximize the expected cumulative rewards in the test task:

$$J(\pi_\theta) = \mathbb{E}_{M \sim p(M)}[J_M(\pi_\theta)]. \tag{3}$$

## 4 Method

### 4.1 Description of Context Shift Problem

OMRL uses pre-collected offline data from task-dependent behavior policy to maximize the expected cumulative reward. Generally, given multiple different tasks, each behavior policy is the standard RL policy trained on each task. Then each offline dataset is collected from the behavior policy interacting with this environment. For $N$ MDPs sampled from $p(P, R)$, the behavior policy of task $M_i$ for

collecting data is $\pi_\beta^i(a|s)$, and the offline data is $D_i = \{(s_{i,j}, a_{i,j}, r_{i,j}, s_{i,j})'\}_{j=1}^{N_{size}}$. The distribution of offline data is denoted by

$$p(D_i) = \rho(s_0) \prod_{j=0}^{N_{size}} \pi_\beta^i(a_j|s_j) R(s_j, a_j) P(s_{j+1}|s_j, a_j). \tag{4}$$

In the meta-test phase, the data is online collected by the exploration policy.

The different sources of offline datasets used in meta-training and online data used in the meta-test lead to the context shift problem: distribution discrepancy between context from behavior policy and context collected from exploration policy. As shown in Eq 4, the distribution of collected offline datasets is jointly determined by task and behavior policy, which means the context used to train the context encoder contains some characteristics specific to the behavior policy. When the characteristic of behavior policy is correlated with the task, the context encoder is possible to mistakenly embed the characteristic of behavior policy as the task information. Since the characteristics of the exploration policy may be very different from the behavior policy, the task representation derived from the context encoder will deliver incorrect task information and cause performance degradation in the meta-test phase.

For example, as shown in Figure 1 left, the task goal is to reach the position on the semicircle, while the offline datasets only contain the trajectories of behavior policy that go towards the corresponding goal. Thus the direction of the behavior policy is its characteristic which is highly correlated with the task goal. Based on this, the context encoder tends to memorize the direction of the context as the task information. During the meta-test phase, the common exploration policy is conditioned on an initial task representation $z_0$ sampled from a prior $p(z)$ to collect context. The initial task representation $z_0$ contains a specific direction, so the collected context will move to this specific direction which is different from behavior policy, as shown in Figure 1 middle. Affected by this context, the agent will go along the specific direction instead of going to the task goal, as shown in Figure 1 right.

To address the context shift problem, we propose that CSRO reduce the effect of policy in the context in both the meta-training and meta-test phases. Figure 2 gives an overview of our CSRO framework.

## 4.2 Max-min Mutual Information Representation Learning

OMRL uses offline datasets to train a context encoder $q_\phi$ to distinguish tasks. We first sample the context $c_i = \{(s_{i,j}, a_{i,j}, r_{i,j}, s'_{i,j})\}_{j=1}^{N_c}$ from the offline datasets $D_i$ of task $M_i$. And then the context encoder extracts each transition embedding from each transition of the context: $z_{i,j} = q_\phi(z|(s_{i,j}, a_{i,j}, r_{i,j}, s'_{i,j}))$. Finally, we use the mean of transition embedding $z_i = \mathbb{E}_j[z_{i,j}]$ as the task representation of task $M_i$. Ideally, the agent should pay more attention to the reward function of the environment where the reward changes, and the dynamic function of the environment where the dynamic changes, and only infer the task from there. However, since the offline dataset is determined by the behavior policy, the task representation will contain the characteristics of the behavior policy. Due to the context shift problem, this context encoder will lead to incorrect task inference in the meta-test phase.

In order to ensure the task representation only focuses on representing the feature of the task , we design the max-min mutual information representation learning mechanism. Since mutual information is a measure of the correlation between two random variables, we can use mutual information to guide the optimization direction of the context encoder. This mechanism consists of two parts, one is to maximize the mutual information between task representation $z$ and the task, and the other is to minimize the mutual information between $z$ and the behavior policy $\pi_\beta$.

Since the task representation needs to represent the task information, we need to maximize the mutual information between task representation and the task. We use distance metric learning in [20] to obtain mutual information approximation with task:

$$L_{maxMI}(\phi) = 1\{y_i = y_j\}\|z_i - z_j\|_2^2 + 1\{y_i \neq y_j\}\frac{\beta}{\|z_i - z_j\|_2^n + \epsilon}, \tag{5}$$

where $c_i$ is context sampled from task $M_i$, $y_i$ is task label, $z_i = q_\phi(z|c_i)$ is a task represtantion. Context encoder minimizes the distance of task representations from the same task and maximizes the distance of task representations from different tasks.

**Definition 1.** In the test task $M_i = (S, A, P_i, \rho, R_i) \sim p(M)$, context $c$ as a context collected by exploration policy $\pi_e$. The expected return of test task $M_i$ which is evaluated by a learned meta-policy $\pi_\theta(a|s, z)$ is:

$$J_{M_i}(\pi_\theta, \pi_e) = E_{s_0 \sim \rho(s_0), z \sim q_\phi(\cdot|c), a_t \sim \pi_\theta(\cdot|s_t, z), r_t \sim R_i(s_t, a_t), s'_t \sim P_i(\cdot|s_t, a_t)}[\sum_{t=0}^{H-1} r_t]. \tag{6}$$

**Proposition 1.** For task $M_i$ and its corresponding behavior policy $\pi_\beta^i$, when exploration policy $\pi_e$ and behavior policy $\pi_\beta^i$ are different, if and only if mutual information between task representations and policies $I(z; \pi_e) = 0$, the expected test return for both is the same $J_{M_i}(\pi_\theta, \pi_e) = J_{M_i}(\pi_\theta, \pi_\beta^i)$.

As analyzed in Section 4.1, the context encoder is possible to contain the characteristics of the behavior policy. Proposition 1 states that we also need to minimize the mutual information between task representations and policies $\pi_\beta$ to remove the influence of behavior policy in task representation. However, due to the collection strategy of offline datasets, each task has only one policy to collect data, which makes it difficult to obtain the representation of behavior policy. Consider that each transition tuple $(s, a, r, s')$ is jointly determined by task and behavior policy $p(a, r, s'|s) = \pi_\beta(a|s)p(s'|s, a)r(s, a)$, we use $(s, a)$ tuple to represent behavior policy. Following [3], we introduce using CLUB to minimize mutual information between transition embedding $z$ and $(s, a)$. Mutual information upper bound estimation of CLUB is defined as:

$$I_{CLUB}(z, (s, a)) = \mathbb{E}_i[\log p(z_i|(s_i, a_i)) - \mathbb{E}_j[\log p(z_j|(s_i, a_i))]]. \tag{7}$$

However the conditional distribution $p(z|(s, a))$ is unavailable on our tasks, therefore we use a variational distribution $q_\psi(z|(s, a))$ to approximate $p(z|(s, a))$ by minimizing Log-likelihood:

$$L_{VD}(\psi) = -\mathbb{E}_{M \sim p(M)}\mathbb{E}_i[\log q_\psi(z_i|(s_i, a_i))]. \tag{8}$$

We minimize mutual information between transition embedding $z$ and $(s, a)$ by minimize mutual information upper bound estimation:

$$L_{minMI}(\phi) = \mathbb{E}_{M \sim p(M)}\mathbb{E}_i[\log q_\psi(z_i|(s_i, a_i)) - \mathbb{E}_j[\log q_\psi(z_j|(s_i, a_i))]], \tag{9}$$

which $z_i$ represents task embedding obtained by $(s_i, a_i, r_i, s'_i)$ through the context encoder. Here, the variational distribution estimator $q_\psi$ and context encoder $q_\phi$ are used for adversarial training. If each transition embedding $z$ obtained by the context encoder $q_\phi(z|(s, a, r, s'))$ contains more $(s, a)$ information, the $q_\psi(z|(s, a))$ has higher accuracy, but mutual information upper bound $L_{minMI}(\phi)$ will be higher. If the transition embedding $z$ does not contain $(s, a)$ information, the ability to predict $z_i$ through $(s_i, a_i)$ is the same as that from other $(s_j, a_j)$, $L_{minMI}(\phi)$ is lower. In this adversarial training, the mutual information between behavior policy and task representation can be minimized, the context encoder will pay more attention to the information of tasks.

By combining the above two mutual information, we can get the total loss of context encoder as:

$$L_{encoder}(\phi) = L_{maxMI} + \lambda L_{minMI}, \tag{10}$$

where $\lambda$ is a hyperparameter to adjust the weight of the two mutual information. Based on the max-min mutual information representation learning, we can reduce the influence of policy in task representation.

### 4.3 Non-prior Context Collection Strategy

In the online meta-test phase, the agent needs to explore the new task to collect context $c$ and update the posterior distribution $q_\phi(z|c)$ according to context $c$. The current exploration strategy of OMRL [27, 4, 26] follows the common exploration strategy of meta-RL [27]. The agent uses the learned meta-policy which is conditioned on initial task representation $z_0$ sampled from the prior distribution $p(z)$ to explore the new task. After that, the agent updates posterior $q_\phi(z|c)$ by collected context $c$ and samples new $z$. Then the agent continues to explore the task to collect context conditioned on $z$. The steps of sampling $z$ and collecting context are performed iteratively. Finally, the agent executes task inference according to all collected contexts.

Because there is only one policy for each task to collect data, the context distribution sampled from offline datasets is highly correlated with the behavior policy. Even if we propose the max-min

mutual information representation learning which minimizes the mutual information between task representation and behavior policy, we still cannot completely eliminate the influence of policy in task representation. If we use the common exploration strategy to collect context $c$, the distribution of context $c$ will correlate with the sampled initial $z_0$. In this way, the posterior $q_\phi(z|c)$ will be affected by $z_0$, leading to incorrect task inference.

On the other hand, there are some methods in current online meta-reinforcement learning that specifically train an exploration policy to collect context [7, 38]. However, in offline settings, there is no way to interact with the environment, and using offline datasets to train the exploration policy, the exploration policy is limited to the vicinity of the datasets, and the conservatism of offline is in conflict with exploration, so by training an exploration policy to solve this problem in offline settings is difficult.

We propose a non-prior context collection strategy to eliminate this problem. Our strategy is not conditioned on sampled initial $z_0$ from prior. The agent first explores the task independently and randomly at each step. Then posterior $p(z|c)$ will be updated after this exploration. Through this exploration, the agent preliminarily recognizes the task. After that, the agent continuously explores the task according to the posterior $q_\phi(z|c)$ and updates the posterior at each step until the context $c$ is collected. In this process, the agent gradually recognizes the task more accurately. Finally, the agent evaluates based on the collected context $c$. By adopting this approach, we mitigate the influence of the sampled initial $z_0$.

### 4.4 Algorithm Summary

Finally, in order to avoid the problem of overestimation of $Q$ caused by the distribution shift of the actions in offline learning, we use BRAC [32] to constrain the similarity between learned policy $\pi_\theta$ and behavior policy $\pi_\beta$ by punishing the distribution shift of learned policy $\pi_\theta$. The optimization objectives under the actor-critic framework become:

$$L_{cirtic}(\omega) = \mathbb{E}_{(s,a,r,s')\sim D, z\sim q_\phi(z|c), a'\sim\pi_\theta(\cdot|s',z)}[(Q_\omega(s,a,z) - r - \gamma Q_\omega^{target}(s',a',z))^2]. \quad (11)$$

$$L_{actor}(\theta) = -\mathbb{E}_{(s,a,r,s')\sim D, z\sim q_\phi(z|c), a''\sim\pi_\theta(\cdot|s,z)}[Q_\omega(s,a'',z) - \alpha D(\pi_\theta(\cdot|s,z), \pi_\beta(\cdot|s,z))]. \quad (12)$$

where $D$ is the divergence estimate of learned meta-policy $\pi_\theta(\cdot|s,z)$ and behavior policy $\pi_\beta(\cdot|s,z)$, and we use the KL divergence.

We summarized our meta-training and meta-test method in the pseudo-code of Appendix A.

## 5 Experiments

We propose max-min mutual information representation learning and the non-prior context collection strategy to mitigate the context shift between the meta-train and meta-test. To demonstrate the performance of our method, we compare CSRO with the prior offline meta-RL methods on meta-RL tasks in which the reward function or dynamic function is variable. Code is available at `https://github.com/MoreanP/CSRO.git`.

### 5.1 Experimental Settings

**Environments.** We evaluate our method on the Point-Robot and MuJoCo [29] that are often used as the offline meta-RL benchmarks:(1) **Point-Robot** is a 2D navigation task of a point in continuous space, and the goal is located on a semicircle far from the origin. (2) **Half-Cheetch-Vel** is to control the agent to reach the target velocity of tasks. (3) **Ant-Goal** is a task modified on Ant, which needs to control the agent to reach different goals on the circle. (4) **Humanoid-Dir** is to control the humanoid agent to move in the direction provided by the task. (5) **Hopper-Rand-Params** is to control the velocity of the agent as fast as possible. (6) **Walker-Rand-Params** is also to control the biped agent to move forward as quickly as possible. For (1) and (3), the goal positions of different tasks are different. (2) and (4) are different in target velocity and direction respectively. These four types of environments are all reward functions changing environments. (5) and (6) are dynamic functions changing environments. Robots in different environments have different physical parameters, including mass, inertia, stamping, and friction.

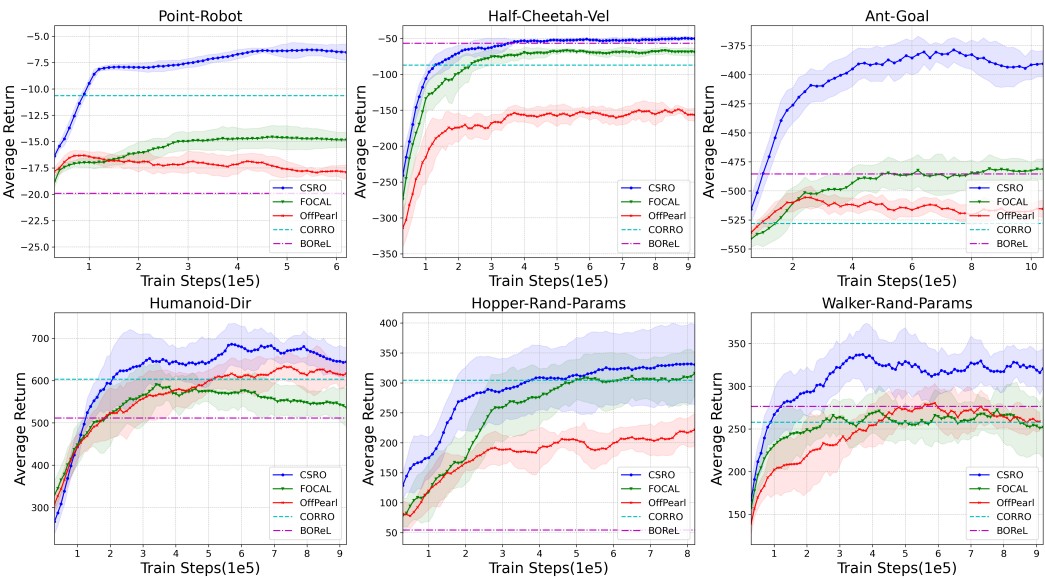

Figure 3: The average return of the online test phase on unseen test tasks versus other OMRL methods. Point-Robot, Half-Cheetah-Vel, Ant-Goal, and Humanoid-Dir are all reward functions changing environments. Hopper-Rand-Params and Walker-Rand-Params are dynamic functions changing environments. The shaded region shows a standard deviation across 8 seeds.

**Offline Data Collections.** For each environment, we sampled 30 training tasks and 10 test tasks from the task distribution. We use SAC [10] on each training task to train an agent and save the policy at different training times as behavior policy. We use each policy to roll out 50 trajectories in the corresponding environment as offline datasets.

**Compared Methods.** To demonstrate the performance of our method, we compare CSRO with the following methods:(1) **OffPearl** is an extension that applies Pearl [27] to offline RL. The context encoder outputs the distribution of task representation and updates by Q-function. The pre-collected offline datasets are used to replace the interaction with the environment. (2) **FOCAL** [20] uses metric distance learning to train context encoder, reduce the task representation distance of the same task, and increase the task representation distance of different tasks. (3) **CORRO** [37] trains CVAE [28] or adds noise to produce negative samples, and uses InfoNCE [22] to train robust context encoder. (4) **BOReL** [4] builds on VariBAD [39]. For a fair comparison, we use a variant of BOReL that does not utilize oracle reward functions, as introduced in the original paper [4].

For fairness comparison, all methods here use the same offline RL algorithm BRAC [32] and the same offline datasets. More details of the experimental setting and results are available in the Appendix.

Table 2: Comparing CSRO with other baselines in the online test phase without and with the non-prior context collection strategy(Np) in Point-Robot, Half-Cheetah-Vel, and Walker-Rand-Params environments.

| Env | Point-Robot | | Half-Cheetah-Vel | | Walker-Rand-Params | |
|---|---|---|---|---|---|---|
| | w/ Np | w/o Np | w/ Np | w/o Np | w/ Np | w/o Np |
| CSRO | **-6.4**±0.8 | -9.2±0.6 | **-48.4**±3.9 | -68.5±13.9 | **344.2**±38.0 | 319.7±38.4 |
| FOCAL | -11.8±1.6 | -14.9±1.1 | -60.9±5.7 | -69.5±9.6 | 253.3±42.7 | 247.5±29.4 |
| OffPearl | -17.0±1.6 | -17.8±1.5 | -133.7±18.9 | -162.8±28.8 | 284.5±30.9 | 262.0±24.5 |
| CORRO | -7.8±1.9 | -10.5±3.0 | -65.6±9.3 | -92.1±23.2 | 312.5±46.6 | 275.2±73.9 |
| BOReL | -21.6±3.9 | -23.2±5.8 | -90.1±28.3 | -56.1±10.7 | 260.6±40.2 | 245.8±32.9 |

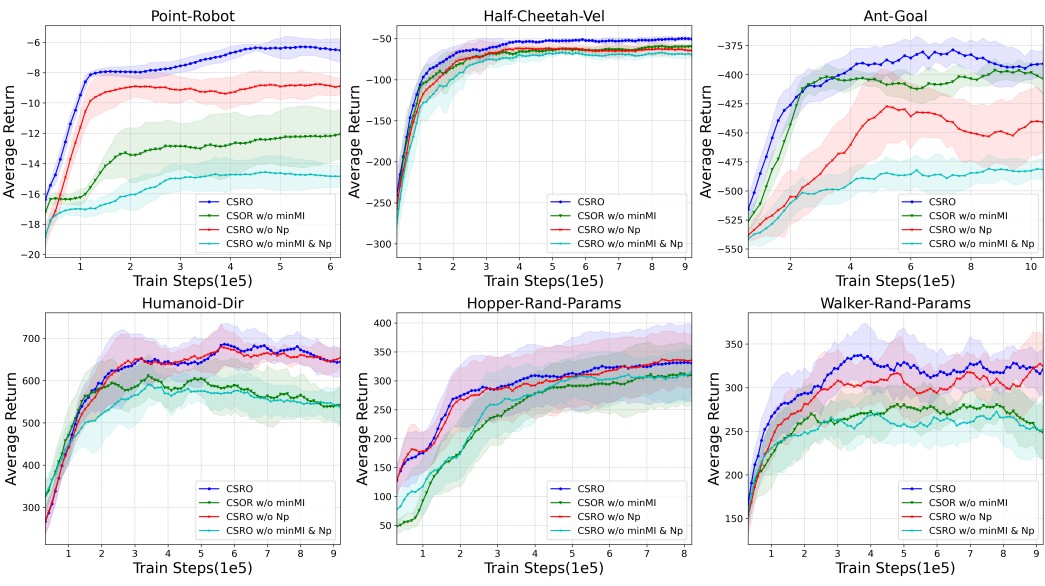

Figure 4: Ablation study in six environments, to compare CSRO with methods that without CLUB minimize mutual information (minMI) and non-prior context collection strategy (Np) components.

## 5.2 Comparison Online Test Results.

To evaluate our performance in the online test phase, we compare CSRO with other methods across all six environments. In the online test phase, each method needs to explore new tasks and collect context. The four baseline methods follow the common exploration strategy, CSRO uses the non-prior context collected strategy that we proposed.

We plot the mean and standard deviation curves of returns across 8 random seeds in Figure 3. In the six environments, CRSO outperforms all the baselines, especially in Point-Robot and Ant-Goal. Experimental results prove the effectiveness of our method.

## 5.3 Ablation.

Max-min mutual information representation learning and non-prior context collection strategy are the key components of our proposed CSRO. Max-min mutual information representation learning uses metric distance learning and CLUB minimizes mutual information. We compare the methods with those without CLUB minimize mutual information(minMI) and non-prior context collection strategy (Np) components, to show the effect of each component. We conducted experiments on six environments, and the results are shown in Figure 4. In six environments, we notice that if both components are not used, the average return of the online test is the lowest. After adding minMI, the average return is improved to varying degrees. In most environments, the average return is also increased after using Np. We noticed that in the Point-Robot and Ant-Goal, our component effect is more significant. Because in the Point-Robot and Ant-Goal, the behavior policy of different tasks moves in different directions, which makes the influence of context shift more serious.

In the online test phase, We also compare the average return of CSRO with other baselines without and with Np. As shown in Table 2, in the online test without Np, CSRO outperforms other baselines in most environments. This shows that CSRO has learned the context encoder with better generalization. Comparing online tests without Np and with Np, the average return of most methods is improved after using Np, this shows the effectiveness of Np. However, the performance of other methods with Np is lower than CSRO, because others do not fully reduce the influence of behavior policy on context encoder.

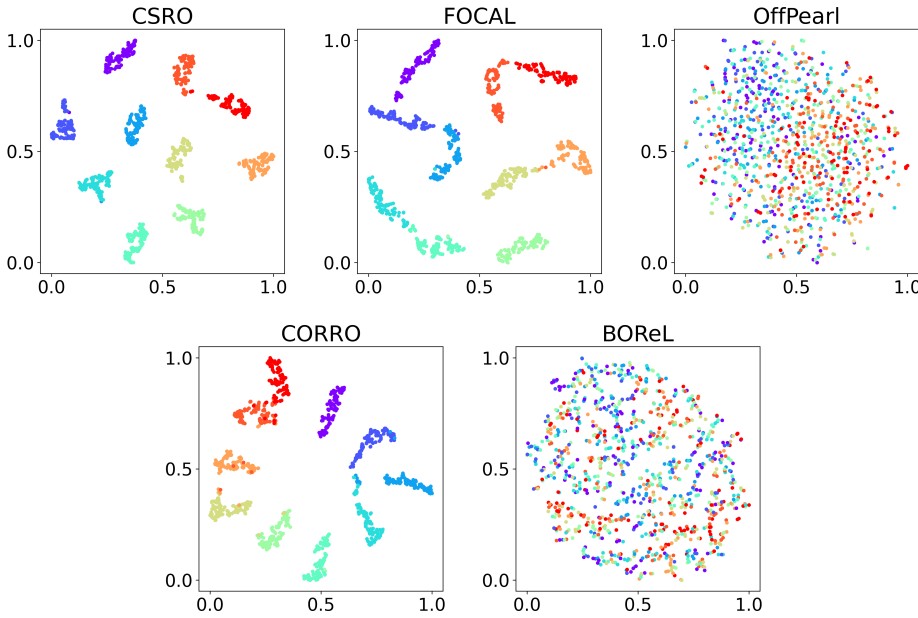

Figure 5: t-SNE visualization of the learned task representation space in Half-Cheetah-Vel. Each point represents a context sampled by the non-prior context collection strategy from test tasks. Tasks of velocities from 1 to 3 are mapped to rainbow colors, from red to purple.

## 5.4 Visualization.

In order to analyze the task representation of the context encoder, we use t-SNE [30] to map the embedding vectors into 2D space and visualize the task representations. For each test task of Half-Cheetah-Vel, we use the non-prior context collection strategy to sample 100 contexts to visualize task embedding. In Figure 5, compared to the other methods, CSRO successfully clusters the task representations of the same task and distinguishes the task representations from different tasks better than other baselines.

## 6 Conclusion

In this paper, we study the context shift problem in OMRL, which arises due to the distribution discrepancy between the context derived from the behavior policy during meta-training and the context obtained from the exploration policy during meta-test. Context shift problem will damage the generalization capability of the meta-policy by leading to incorrect task inference on unseen tasks. We propose that CSRO address the context shift problem with only offline datasets with the key insight of reducing the effect of policy in context during both meta-training and meta-test. In meta-training, we design the max-min mutual information representation learning mechanism which reduces the influence of behavior policy in task representation. In the meta-test, we introduce the non-prior context collection strategy to reduce the effect of the exploration policy. We compared CSRO with previous OMRL methods on a range of challenging domains with reward or dynamic function changes. Experimental results show that CSRO can substantially reduce the context shift and outperform prior methods.

## Acknowledgments

This work is partially supported by the NSF of China(under Grants 61925208, 62102399, 62222214, 62002338, U22A2028, U19B2019), Beijing Academy of Artificial Intelligence (BAAI), CAS Project for Young Scientists in Basic Research(YSBR-029), Youth Innovation Promotion Association CAS and Xplore Prize.

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

# A   Pseudo-Code

---

**Algorithm 1** CSRO Meta-training

---

**Input**: Offline Datasets $D = \{D_i\}_{i=1}^{N_{env}}$ of a set of training tasks $\{M_i\}_{i=1}^{N_{env}}$, initialize learned policy $\pi_\theta$, Q-function $Q_\omega$, context encoder $q_\phi$, and CLUB encoder $q_\psi$, hyperparameter $\lambda$
**Parameter**: $\theta, \omega, \phi, \psi$

1: **while** not done **do**
2:     **for** step in training steps **do**
3:         Sample buffer $D_i \sim D$ and context from buffer $c = \{(s_j, a_j, r_j, s_j')\} \sim D_i$, history transitions $h \sim D_i$.
4:         Compute each transition embedding $z = q_\phi(z|(s, a, r, s'))$, $z = q_\psi(z|(s, a))$ and task representation $z = q_\phi(z|c)$
5:         Compute $L_{VD}(\psi)$
6:         Update $\psi$ to minimize $L_{VD}(\psi)$
7:         Compute $L_{encoder}(\phi) = L_{maxMI}(\phi) + \lambda L_{minMI}(\phi)$
8:         Update $\phi$ to minimize $L_{encoder}(\phi)$
9:         Use history transitions $h$ to compute $L_{critic}(\omega)$, $L_{actor}(\theta)$
10:        Update $\theta, \omega$ to minimize $L_{critic}(\omega)$, $L_{actor}(\theta)$
11:     **end for**
12: **end while**

---

---

**Algorithm 2** CSRO Meta-testing

---

**Input**: A set of testing tasks $\{M_i\}_{i=1}^{N_{env}}$, learned policy $\pi_\theta$, context encoder $q_\phi$, random explore step $t_r$

1: **for** each task $M_i$ **do**
2:     $c = \{\}$
3:     **for** $t = 0, \ldots, T - 1$ **do**
4:         **if** $t < t_r$ **then**
5:             Agent samples a random action $a_t$ to roll out $(s_t, a_t, r_t, s_t')$
6:         **else**
7:             Compute posterior $z = q_\phi(z|c)$.
8:             Agent use $\pi_\theta(a|s, z)$ roll out $(s_t, a_t, r_t, s_t')$
9:         **end if**
10:        $c = c \cup (s_t, a_t, r_t, s_t')$
11:     **end for**
12:     Compute posterior $z = q_\phi(z|c)$.
13:     Roll out policy $\pi_\theta(a|s, z)$ for evaluation
14: **end for**

---

# B   Proof of Proposition 1

**Proposition 1.** For task $M_i$ and its corresponding behavior policy $\pi_\beta^i$, when exploration policy $\pi_e$ and behavior policy $\pi_\beta^i$ are different, if and only if mutual information between task representations and policies $I(z; \pi_e) = 0$, the expected test return for both is the same $J_{M_i}(\pi_\theta, \pi_e) = J_{M_i}(\pi_\theta, \pi_\beta^i)$.

*Proof.* The meta-policy gets the highest expected return when exploring policy and behavior policy are the same: $\pi_e = \pi_\beta^i$. However, for any exploration policy $\pi_e$, we hope $J_{M_i}(\pi_\theta, \pi_e) = J_{M_i}(\pi_\theta, \pi_\beta^i)$, if and only if for any $\pi_e$, $q_\phi(z|c)$ is the same. Since $c$ is collected by $M_i$ and $\pi_e$, $q_\phi(z|c)$ can be written as $q_\phi(z|M_i, \pi_e)$, i.e. for any $\pi_e$, $q_\phi(z|M_i, p_e)$ is the same. So $z$ and $\pi_e$ are independent, that is, the mutual information $I(z; \pi_e) = 0$.

# C   Environment Details

In this section, we show details about the environments of our experiment.

**Point-Robot:** A problem of control point robot navigation in 2D space. The start position is fixed to $(0,0)$. The goal of each task is located on a unit semicircle centered on the start position. Each task needs to control the robot from the start position to the goal. The state space is $\mathbb{R}^2$, comprising the XY position of the robot. The action space is $[-1,-1]^2$, with each dimension corresponding to the moving distance in the XY direction. The reward function is defined as the negative distance from the goal.

**Half-Cheetah-Vel:** Control a Cheetah to move forward and achieve goal velocity. The target velocity is sampled from $[1,3]$. The state space is $\mathbb{R}^{20}$, comprising the position and velocity of the cheetah; the angle and angular velocity of each joint. The action space is $[-1,1]^6$, with each dimension corresponding to the torque of each joint. The reward function is the absolute difference between the agent's velocity and the target velocity plus the control cost.

**Ant-Goal:** The Ant-Goal task consists of controlling an "ant" robot to navigate. The goal of each task is located on a circle with radius 2 centered on $(0,0)$. The state space is $\mathbb{R}^{29}$, comprising the position and velocity of the ant as well as the angle and angular velocity of 8 joints. The action space is $[-1,1]^8$, with each dimension corresponding to the torque of each joint. The reward function is defined as the negative distance from the goal plus the control cost.

**Humanoid-Dir:** The Humanoid-Dir task consists of controlling a "humanoid" robot in the target direction. The target direction of each task is sampled from $[0,2\pi]$. The state space is $\mathbb{R}^{376}$ and the action space is $[-1,1]^{17}$. The reward function is the dot between the velocity of the robot and the target direction plus the staying alive bonus and control cost.

**Hopper-Rand-Params:** The Hopper-Rand-Params controls a one-legged robot to move forward. The source code is taken from the rand_param_envs repository. [2] The tasks are varied in body mass, body inertia, joint damping, and friction. Each parameter is the product of the default value and the coefficient sampled from $[1.5^{-3}, 1.5^3]$. The state space is $\mathbb{R}^{11}$ and the action space is $[-1,1]^3$. The reward function is forward velocity plus the staying alive bonus and control cost.

**Walker-Rand-Paras:** The Walker-Rand-Params controls a bi-pedal robot to move forward, also from the rand_param_envs repository. Each parameter is obtained in the same way as Hopper-Rand-Params and the reward function is the same as Hopper-Rand-Params. The state space is $\mathbb{R}^{17}$ and the action space is $[-1,1]^6$.

# D Offline Data Collections

For each task, we sample 40 environments from environment distribution. Out of these, 30 environments are designated as training environments, while the remaining 10 environments serve as test environments. We employ SAC [10] to train an agent on each training environment and save the policy at different training steps. To create offline datasets, we generate 50 trajectories using each policy from every environment. Table 3 presents the hyperparameters employed during the collection of offline datasets.

Table 3: Hyperparameters used in offline datasets collection.

| Hyperparameters | Point-Robot | Half-Cheetah-Vel | Ant-Dir | Humanoid-Dir | Hopper-Rand-Params | Walker-Rand-Params |
|---|---|---|---|---|---|---|
| Training steps | 5000 | 1e6 | 1e6 | 1e6 | 1e6 | 1e6 |
| Initial steps | 2e3 | 5e4 | 5e4 | 5e4 | 5e4 | 5e4 |
| Eval frequency | 200 | 5e4 | 5e4 | 5e4 | 5e4 | 5e4 |
| Sampling episodes | 50 | 50 | 50 | 50 | 50 | 50 |
| Learning rate | 1e-4 | 1e-4 | 1e-4 | 1e-4 | 1e-4 | 1e-4 |
| Batch size | 1024 | 1024 | 1024 | 1024 | 1024 | 1024 |

# E Experimental Setting

For each task, We use offline datasets collected at different times to train. Details of using offline datasets in Table 4:

We list other hyperparameters in the offline meta-training phase in Table 5.

---

[2]https://github.com/dennisl88/rand_param_envs.

Table 4: Details of using offline datasets: The 'Checkpoints' column indicates the data collected by policies at different steps during the meta-training phase. The three numbers denote the starting steps, ending steps, and step spacing.

| Env | Checkpoints |
|---|---|
| Point-Robot | [2200, 4800, 200] |
| Half-Cheetah-Val | [100000, 950000, 50000] |
| Ant-Goal | [100000, 950000, 50000] |
| Humanoid-Dir | [50000, 950000, 50000] |
| Hopper-Rand-Params | [50000, 950000, 50000] |
| Walker-Rand-Params | [50000, 950000, 50000] |

Table 5: Hyperparameters used in offline meta-training.

| Hyperparameters | Point-Robot | Half-Cheetah-Vel | Ant-Dir | Humanoid-Dir | Hopper-Rand-Params | Walker-Rand-Params |
|---|---|---|---|---|---|---|
| Reward scale | 100 | 5 | 5 | 5 | 5 | 5 |
| Latent dimension | 20 | 20 | 20 | 20 | 40 | 40 |
| Use BRAC | False | True | True | True | True | True |
| Batch size | 256 | 256 | 256 | 256 | 256 | 256 |
| Meta batch size | 16 | 16 | 10 | 16 | 16 | 16 |
| Embedding batch size | 1024 | 100 | 512 | 256 | 256 | 256 |
| Actor Learning rate | 3e-4 | 3e-4 | 3e-4 | 3e-4 | 3e-4 | 3e-4 |
| Critic Learning rate | 3e-4 | 3e-4 | 3e-4 | 3e-4 | 3e-4 | 3e-4 |
| Encoder Learning rate | 3e-4 | 3e-4 | 3e-4 | 3e-4 | 3e-4 | 3e-4 |
| Maximum episode length | 20 | 200 | 200 | 200 | 200 | 200 |
| MinMI loss weight $\lambda$ | 25 | 10 | 50 | 50 | 25 | 25 |
| behavior regularization | 50 | 50 | 50 | 50 | 50 | 50 |
| Discount factor | 0.9 | 0.99 | 0.99 | 0.99 | 0.99 | 0.99 |

# F  Additional Experiments

## F.1  Comparison Offline Test Results

Offline testing is an ideal evaluation method where the context used is sampled from the pre-collected offline data in the test task, thereby disregarding the context shift problem. To assess our performance in the offline test phase, we adopt the same approach as the training environment and collect offline datasets as context on the testing environment.

We compare CSRO with other methods across all six environments and plot the mean and standard deviation curves of returns based on 8 random seeds in Figure 6. In most environments, CSRO demonstrates competitive performance compared to other baselines. Experimental results demonstrate the effectiveness of our algorithm, even in the absence of addressing the context shift problem.

## F.2  Ablation Offline and Online Test

Under the offline test scenario that ignores the context shift problem, the algorithm can achieve its highest performance. We compare the performance of the online testing method that uses the non-prior context collection strategy(Np) with offline testing. We conduct experiments on six environments and plot the mean and standard deviation curves of returns across 8 random seeds in Figure 7.

In most environments, the performance of CSRO that uses Np is close to the offline test. There exists a gap between Point-Robot and Ant-Goal environments due to the particularly severe and challenging context shift problem in these two environments. However, our approach still outperforms previous methods. The experimental results highlight the efficacy of our approach in addressing the context shift issue, albeit with some remaining challenges in these specific environments.

## F.3  Visualize Contexts and Trajectory of Online Test

Lastly, we further study the non-prior context collection strategy. Figure 8 showcases two different tasks in the Point-Robot environment. We illustrate the context gathered through the non-prior context collection strategy and the corresponding trajectory navigation. Observing the visualizations, we notice that the agent's perception of the task improves after random exploration. Subsequent explorations enhance the agent's understanding of the environment, enabling it to accurately navigate toward the goal.

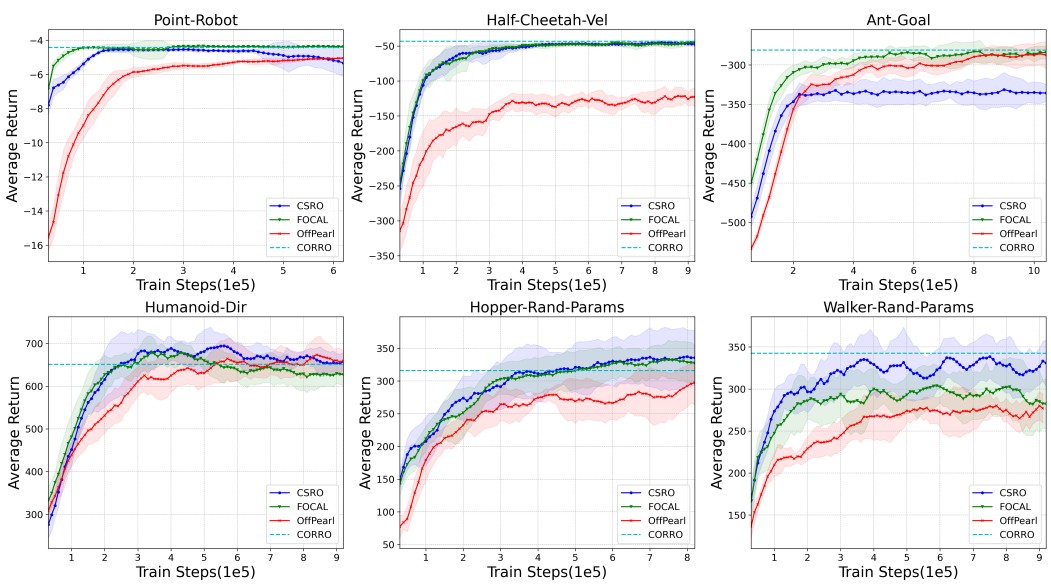

Figure 6: Compared to other OMRL methods, CSRO's offline testing averages returns on unseen testing tasks

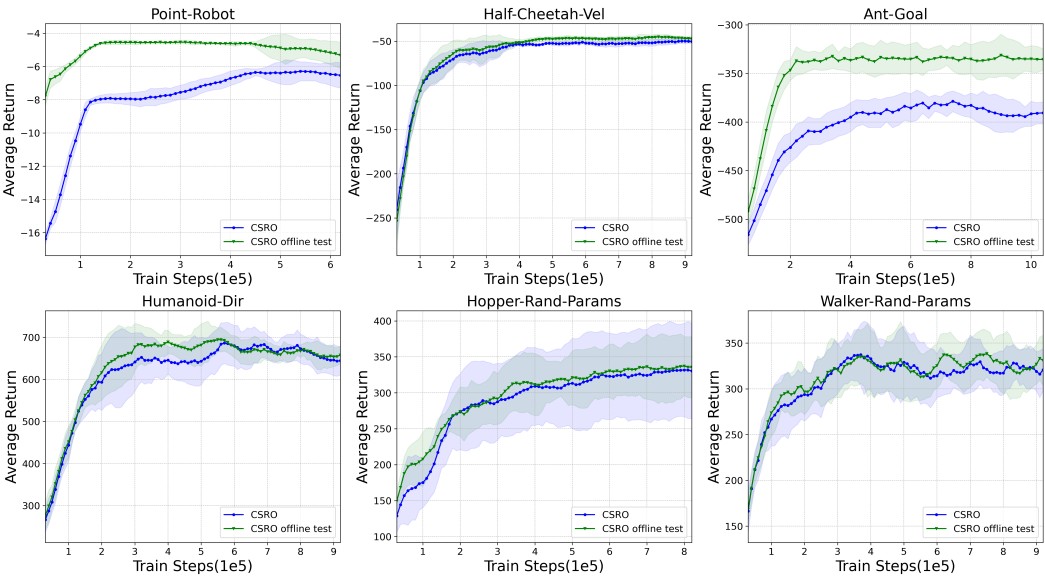

Figure 7: The average return of the offline test and the online test that uses the non-prior context collect strategy on unseen test tasks.

## F.4 Additional Visualization

We have added visualizations of task representations for CSRO, FOCAL [20], and CORRO [37] in the Point-Robot environment, as shown in Figure 9. We can see that FOCAL is worse. In CORRO, similar tasks are closer, and many points of the same task are far apart.

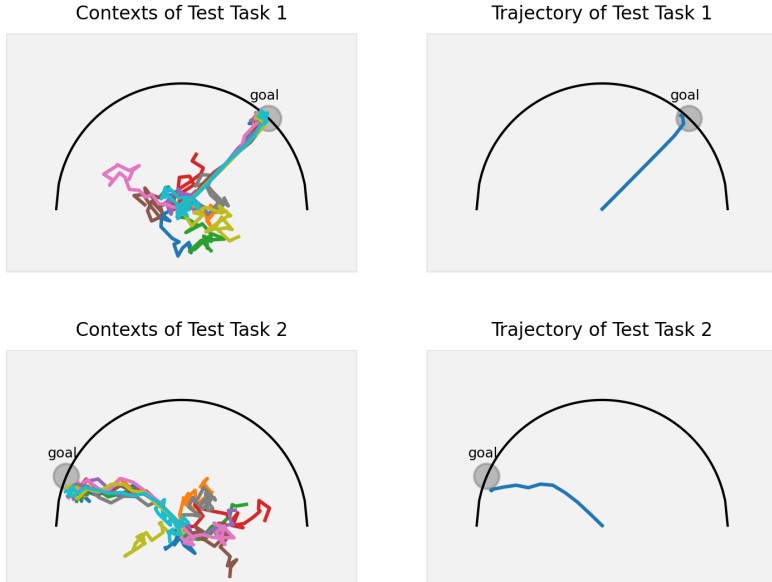

Figure 8: Visualization of contexts and trajectory after using the non-prior context collection strategy in the Point-Robot environment.

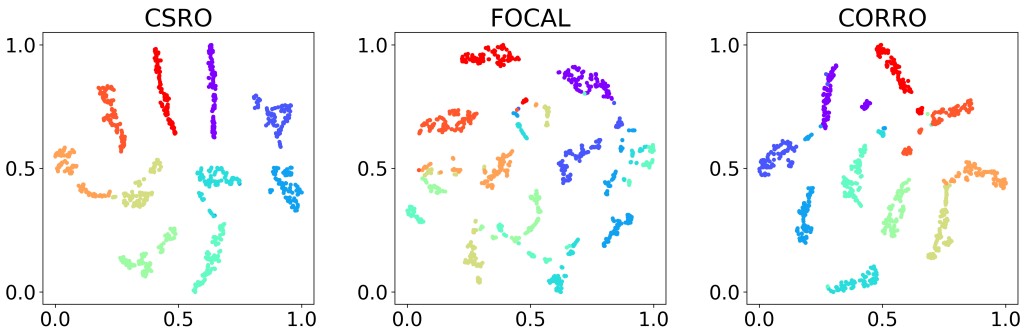

Figure 9: t-SNE visualization of the learned task representation space in Point-Robot. Each point represents a context sampled by the non-prior context collection strategy from test tasks. The direction of the goal of the task is from 0 degrees to 180 degrees, from red to purple.

## F.5 Additional Exploration Policy

In Section 4.3, we discussed how current online meta-RL attempts to train an exploration policy for collecting context, an approach that isn't suitable for addressing the context shift problem in offline meta-RL. Here, we made an attempt to train an exploration policy in MetaCURE [38] using offline data, and the experimental results are presented in Table 6.

Observably, there is a substantial drop in experimental performance when using MetaCURE's exploration policy. This is because, in an offline setting, the exploration policy cannot explore the environment, and training the exploration policy with offline data confines it to the vicinity of that data. Consequently, during testing, the exploration policy becomes ineffective when encountering previously unobserved states.

Table 6: Training MetaCURE's exploration policy with offline datasets for the purpose of collecting context during testing.

| Env | Point-Robot | Half-Cheetah-Vel |
|---|---|---|
| CSRO | **-6.4**±0.8 | **-48.4**±3.9 |
| CSRO+MetaCURE | -13.2±2.3 | -87.9±8.9 |

## F.6 Compare with IDAQ

We compare our work with IDAQ [31], which is a recent study addressing the context shift issue. IDAQ uses the same training approach as FOCAL, but it tackles this problem by attempting to collect context during testing that closely matches the distribution seen during training. As depicted in Table 7, the experimental results reveal that the performance of CSRO is comparable to that of IDAQ. CSRO exhibits slightly lower performance on the Point-Robot compared to IDAQ, but it significantly outperforms IDAQ on the Half-Cheetah-Vel and Walker-Rand-Params tasks.

Table 7: CSRO and IDAQ compare performance on three environments.

| Env | Point-Robot | Half-Cheetah-Vel | Walker-Rand-Params |
|---|---|---|---|
| CSRO | -6.4±0.8 | **-48.4**±3.9 | **344.2**±38.0 |
| IDAQ | **-5.2**±0.1 | -60.9±6.5 | 297.0±23.6 |

Furthermore, we also compared the performance of several methods in sparse settings, as shown in Table 8. It can be observed that, except for IDAQ, all other methods yield poor results. For CSRO, with a non-prior context collection strategy, it becomes challenging to obtain context with rewards and infer the environment, rendering it ineffective.

Table 8: Compared the performance of several methods in Sparse-Point-Robot.

| Method | CSRO | FOCAL | OffPearl | IDAQ |
|---|---|---|---|---|
| Sparse-Point-Robot | 0.8±0.1 | 0.8±0.1 | 0.6±0.1 | **6.5**±0.3 |

We conduct a detailed analysis of IDAQ's performance in the Sparse-Point-Robot. First, we introduce the configuration of the Sparse-Point-Robot environment: The task's goals are randomly distributed along a semicircular arc with a radius of 1 from the starting point. We select 30 goals from this distribution for training tasks and 10 goals for testing tasks. The agent only receives a non-zero reward when its distance to the goal is less than 0.2. This implies that, on average, there are four training goals within a radius of 0.2 for each tested goal.

Next, we will analyze IDAQ in Sparse-Point-Robot: IDAQ retains all the $z$ from the training tasks and maintains a used $z$-set during the testing phase. During the prior exploration phase, IDAQ samples a $z$ from the training tasks which has the maximum distance from the used $z$-set (the difference between the direction of movement of sampled $z$ and used $z$-set is the largest, and adds it to the used $z$-set) to gather context. If the accumulated reward from the collected context surpasses that of all previously collected contexts, the current context is retained; otherwise, it is discarded. IDAQ samples the $z$ with the maximum distance a total of 10 times.

This signifies that the agent explores 10 directions of training goals which have large differences in direction, effectively covering a substantial portion of the entire space. Considering that there are 4 training goals near each test goal, this implies that the sampling can capture the vicinity of the test goal, enabling the agent to attain a non-zero reward. By retaining only high-reward contexts, this indicates that after the prior exploration phase, the remaining contexts exhibit a small context shift and offer non-zero rewards (containing environmental information). As a result, during the subsequent posterior sampling, IDAQ can update its belief and make improved inferences about the testing task.

Based on the aforementioned analysis, we can identify the key environmental factors under which IDAQ operates effectively in the Sparse-Point-Robot: presence of training goals within a radius of 0.2 from the testing goal is crucial, without this condition, the prior exploration process may encounter difficulty in collecting rewarding contexts. To validate this hypothesis, we modified the configuration of the Sparse-Point-Robot environment, resulting in three distinct setups: Setting A reduces the radius

for obtaining rewards from 0.2 to 0.05; Setting B removes the four training goals near the testing goal. Setting C introduces out-of-distribution (OOD) tasks, where training goals are located within the arc of $[0, \frac{3}{4}\pi)$, while testing goals are situated in the interval $[\frac{3}{4}\pi, \pi]$. In these three setups, there are no training goals within the reward radius of the testing goals. We conducted performance testing of IDAQ under these three setups, as shown in Table 9 above. Significant performance degradation can be observed, confirming our conjecture.

Table 9: Comparing IDAQ's performance on the Sparse-Point-Robot under various settings.

|  | Sparse-Point-Robot | Setting A | Setting B | Setting C |
|---|---|---|---|---|
| IDAQ | **6.5**±0.3 | 0.4±0.1 | 1.1±0.8 | 1.3±0.2 |

Hence, we can conclude that for sparse environments, IDAQ is suitable when both the training tasks and testing tasks follow independent and identically distributed (IID). In addition, it is essential for the reward range of testing tasks to encompass certain training tasks.

Overall, both CSRO and IDAQ aim to address the issue of context shift in offline meta RL. CSRO's goal is to acquire a more essential context encoder and reduce the shift during prior exploration. IDAQ, on the other hand, aims to collect a context during the testing phase that shares the same distribution as that of the training phase. Hence, both methods exhibit certain limitations. For instance, CSRO exhibits poor performance in sparse environments; IDAQ needs more exploration steps and has limitations in its applicability to certain sparse environments.

