# OpenReview forum: "Context Shift Reduction for Offline Meta-Reinforcement Learning"
_NeurIPS.cc/2023/Conference — NeurIPS 2023 poster_

### Official Review · Reviewer_qvwn · 2023-06-23

**Soundness:** 3 good
**Presentation:** 3 good
**Contribution:** 3 good
**Rating:** 7
**Confidence:** 4

**Summary:**

This paper proposes CSRO, an offline Meta-RL algorithm that deals with the distributional shift problem in offline meta-RL with online adaptation. CSRO addresses this problem by constraining the task encoding to only contain information about transition and reward, but not contain information about state-action distribution. CSRO also proposes to use random exploration at the start of exploration to further address the distribution mismatch problem. Experiment results show improved performance on MuJoCo task sets.

**Strengths:**

1. The distributional shift problem is a fundamental problem in offline meta-RL with online adaptation.
2. The information-theoretic regularization on task embeddings is novel and interesting.
3. Presentation is clear, and the paper is easy to follow.

**Weaknesses:**

1. The evaluation tasks are a bit too simple. I expect the authors to evaluate on more complex task distributions like Meta-World ML1, which is more challenging and convincing.
2. I am concerned about the efficiency of the proposed exploration method. Random exploration can be very ineffective and may struggle on hard tasks like Meta-World or sparse reward tasks.

**Questions:**

1. Can the authors evaluate CSRO on Meta-World ML1? These tasks are more challenging and convincing.
2. There is a recent work that also addresses the problem of offline meta-RL with online adaptation [1]. Although this work is contemporary to CSRO and I do not require the authors to compare these two algorithms empirically, I expect the authors to discuss pros and cons of CSRO compared to [1].
3. Is random exploration a reasonable choice? Will it fail on more complex or sparse-reward tasks?






[1] Offline Meta Reinforcement Learning with In-Distribution Online Adaptation. https://openreview.net/forum?id=dkYfm01yQp

**Limitations:**

I encourage the authors to add some discussion on CSRO in the paper. I think the random exploration is one important limitation.

---

> ### Author Rebuttal · Authors · 2023-08-09
>
> Thanks for your detailed review. We are glad to discuss your concerns one by one.
>
> > **Q1**: Can the authors evaluate CSRO on Meta-World ML1? These tasks are more challenging and convincing.
>
> **A1**: We conduct experiments on meta-world ML1, and we can see that CSRO has higher performance than FOCAL, which proves that our method is effective.
>
> | Env   | Reach-v2 |
> | ----- | -------- |
> | CSRO  | 0.19     |
> | FOCAL | 0.10     |
>
>
>
> > **Q2**: There is a recent work that also addresses the problem of offline meta-RL with online adaptation [1]. Although this work is contemporary to CSRO and I do not require the authors to compare these two algorithms empirically, I expect the authors to discuss pros and cons of CSRO compared to [1].
>
> **A2**:We reproduced the performance of IDAQ on the three environments and found that CSRO and IDAQ are comparable.
>
> | Env  | Point-Robot | Half-Cheetah-vel | Walker-Rand-Params |
> | ---- | ----------- | ---------------- | ------------------ |
> | CSRO | -6.4        | -48.4            | 344.2              |
> | IDAQ | -5.2        | -60.9            | 297.0              |
>
> Because the IDAQ collection context tends to choose a context with a higher reward, its performance in the expert datasets should be better. However, due to its approximate greedy iteration, if the number of adaptation steps is reduced, the performance will drop. We found that IDAQ on Half-Cheetah-Vel, adapting to 1000 steps drops to using 600 steps, and the performance will drop from -60.9 to -90.1, while CSRO uses 600 steps and performance is  -48.4.
>
>
>
> > **Q3**: Is random exploration a reasonable choice? Will it fail on more complex or sparse-reward tasks?
>
> A3: We described in Figure 1 and Section 4.3 that the common exploration method collects context, and the collected context will be related to the sampled initial $z_{0}$. Subsequent posterior $q_{\phi}(z|c)$ are also affected by $z_{0}$, leading to wrong task inferences. Therefore, we propose that we need to eliminate this kind of erroneous prior, first, use a small amount of random data for initial context collection and then meta-policy $\pi_{\theta}(a|s,z)$ continues to explore and collect context. Since it is infeasible to train an exploration strategy like online meta RL using only offline data, our approach is reasonable.
>
> The CORRO and OffPearl methods we compared were also not tested in sparse environments, and our method may not work in sparse environments.

---

> > ### Comment · Reviewer_qvwn · 2023-08-11
> > **Thank you for your response**
> >
> > I thank the authors for their efforts and their response has largely addressed my concerns. I extremely appreciate the authors for testing IDAQ within such limited time. I have a further question:
> >
> > For Q3, a benefit of posterior sampling is that it has some potential ability to deal with some reward sparsity, as it iteratively updates its belief and explores the environment. As shown in Figure 4. in [1], In sparse reward environments, PEARL's posterior sampling will actively explore possible goals and update its belief. E.g., if it explores a potential goal region and finds that it is not the real goal, it will update its task belief to exclude that explored goal. This mechanism makes PEARL (also IDAQ and OffPEARL)'s exploration possibly more efficient than CSRO. This might be a limitation of CSRO, as CSRO discards posterior sampling to some extent (to fix context distribution shift) and may harm exploration efficiency as well as performance on sparse reward tasks.
> >
> > I would like to raise my score if the authors add a discussion on this limitation (E.g., conduct experiments in simple sparse reward environments like Figure 4. in [1]) and add this discussion in the final version of the paper.
> >
> > [1] Rakelly, Kate, et al. "Efficient off-policy meta-reinforcement learning via probabilistic context variables." International conference on machine learning. PMLR, 2019.

---

> > > ### Author Response · Authors · 2023-08-18
> > >
> > > Thank you very much for your response. We will further discuss your question.
> > >
> > > Firstly, we would like to clarify a viewpoint: As mentioned in lines 219-226 of the paper, CSRO also incorporates posterior sampling. Both CSRO and PEARL follow a similar process of initially collecting data from the environment, iteratively updating the posterior, and then utilizing the updated posterior for environment exploration.  The distinction is: PEARL initially employs $z\sim p(z)$ to explore, and CSRO initially employs random exploration to collect data.
> > >
> > > We evaluated the performance of CSRO, FOCAL, and OffPEARL in the sparse-point-robot environment, as depicted in the following table:
> > >
> > > | CSRO | FOCAL | OffPEARL |
> > > | ---- | ----- | -------- |
> > > | 0.76 | 0.78  | 0.61     |
> > >
> > > We can observe that all three methods exhibit poor performance, significantly below the performance of the expert policy at 10.6.
> > >
> > > For the sparse-point-robot environment, although PEARL can effectively explore the testing environment during online RL, OffPEARL cannot recognize the testing environment during offline RL.
> > >
> > > In the online RL, the policy for collecting context during the testing phase remains consistent with that used during the training phase. For the sparse-point-robot environment, whether it's prior or posterior exploration, PEARL's agent moves in some directions, potentially without receiving rewards, which makes it unable to directly infer the specific environment. However, it can eliminate certain environments from consideration, so it contains some useful environmental information. Furthermore, PEARL encounters similar exploratory behaviors during the training phase empowers it to effectively use contextual environmental information. This allows it to continuously refine its beliefs throughout the entire iteration process.
> > >
> > > In the offline RL, the training phase's context solely originates from the behavior policy $\mu$, while this is not the case during the testing phase. In the sparse-point-robot environment, even though PEARL's exploration process collects context with partial environmental information, but the collected contexts exhibit significant distributional shifts and remain previously unseen. This hinders PEARL's ability to effectively utilize environmental information and update its belief accurately, resulting in poor performance.
> > >
> > > Our method CSRO is primarily suited for dense environments with the aim of reducing context shift between training and testing phases. We don't incorporate additional design for sparse environments. In the sparse-point-robot environments, even though CSRO can collect contexts with minor distributional shifts, it encounters difficulty in capturing meaningful environmental information. As a result, its performance is also poor. In the sparse environments of offline RL,  addressing this issue necessitates the simultaneous collection of contexts with minimal distributional shifts and containing pertinent environmental information.
> > >
> > > We will subsequently include a discussion in the paper about the limitations of our method in the sparse environments of offline RL.

---

> > > > ### Comment · Reviewer_qvwn · 2023-08-19
> > > >
> > > > I thank the authors for their efforts and I largely agree with them. I would like to mention that a sparse-reward comparison with FOCAL and OFFPEARL is not fair enough, as they are originally designed for adaptation with offline data. Comparison with IDAQ is the most suitable, as it exactly solves the distribution shift problem in offline meta-RL with online adaptation, as the authors mention: "the collected contexts exhibit significant distributional shifts and remain previously unseen. This hinders PEARL's ability to effectively utilize environmental information and update its belief accurately, resulting in poor performance." I am currently maintaining my score for this little flaw in the author's response, but overall I am positive about this paper. I will raise my score if the authors add discussions on IDAQ in this sparse-reward issue.

---

> > > > > ### Author Response · Authors · 2023-08-19
> > > > >
> > > > > Thank you very much for your response. We will continue to discuss the situation of IDAQ in sparse environments.
> > > > >
> > > > > We have evaluated the performance of IDAQ in a sparse-point-robot environment. As shown in the table below, IDAQ achieves a performance of 6.50 compared to the expert policy's 10.6. This performance surpasses that of several previous methods.
> > > > >
> > > > > |      | sparse-point-robot | Setting A | Setting B | Setting C |
> > > > > | ---- | ------------------ | --------- | --------- | --------- |
> > > > > | IDAQ | 6.50               | 0.37      | 1.08      | 1.33      |
> > > > >
> > > > > First, we introduce the configuration of the sparse-point-robot environment: The task's goals are randomly distributed along a semicircular arc with a radius of 1 from the starting point. We select 30 goals from this distribution for training tasks and 10 goals for testing tasks. The agent only receives a non-zero reward when its distance to the goal is less than 0.2. This implies that, on average, there are four training goals within a radius of 0.2 for each tested goal.
> > > > >
> > > > > Next, we will analyze IDAQ in sparse-point-robot:
> > > > >
> > > > > IDAQ retains all the $z$ from the training tasks and maintains a used $z$-set during testing phase. During the prior exploration phase, IDAQ samples a $z$ from the training tasks which has the maximum distance from the used $z$-set (the difference between the direction of movement of sampled $z$ and used $z$-set is the largest, and adds it to the used $z$-set) to gather context. If the accumulated reward from the collected context surpasses that of all previously collected contexts, the current context is retained; otherwise, it is discarded. IDAQ samples the $z$ with the maximum distance a total of 10 times.
> > > > >
> > > > > This signifies that the agent explores 10 directions of training goals which have large differences in direction, effectively covering a substantial portion of the entire space. Considering that there are 4 training goals near each test goal, this implies that the sampling can capture the vicinity of the test goal, enabling the agent to attain a non-zero reward. By retaining only high-reward contexts, this indicates that after the prior exploration phase, the remaining contexts exhibit a small context shift and offer non-zero rewards (containing environmental information). As a result, during the subsequent posterior sampling, IDAQ can update its belief and make improved inferences about the testing task.
> > > > >
> > > > > Based on the aforementioned analysis, we can identify the key environmental factors under which IDAQ operates effectively in the sparse-point-robot: Presence of training goals within a radius of 0.2 from the testing goal is crucial, without this condition, the prior exploration process may encounter difficulty in collecting rewarding contexts. To validate this hypothesis, we modified the configuration of the sparse-point-robot environment, resulting in three distinct setups: Setting A reduces the radius for obtaining rewards from 0.2 to 0.05; Setting B removes the four training goals near the testing goal. Setting C introduces out-of-distribution (OOD) tasks, where training goals are located within the arc of $[0,\frac{3}{4}\pi)$, while testing goals are situated in the interval $[\frac{3}{4}\pi,\pi]$. In these three setups, there are no training goals within the reward radius of the testing goals. We conducted performance testing of IDAQ under these three setups, as shown in the table above. Significant performance degradation can be observed, confirming our conjecture.
> > > > >
> > > > > Hence, we can conclude that for sparse environments, IDAQ is suitable when both the training tasks and testing tasks follow independent and identically distributed (IID). In addition, it is essential for the reward range of testing tasks to encompass certain training tasks.
> > > > >
> > > > > Overall, both CSRO and IDAQ aim to address the issue of context shift in offline meta RL. CSRO's goal is to acquire a more essential context encoder and reduce the shift during prior exploration. IDAQ, on the other hand, aims to collect a context during the testing phase that shares the same distribution as that of the training phase. Hence, both methods exhibit certain limitations. For instance, CSRO exhibits poor performance in sparse environments; IDAQ needs more exploration steps and has limitations in its applicability to certain sparse environments. We will subsequently include this discussion in the paper.

---

> > > > > > ### Comment · Reviewer_qvwn · 2023-08-20
> > > > > >
> > > > > > I thank the authors for their reply and appreciate their discussion. I am increasing my score from 6 to 7.

---

### Official Review · Reviewer_DbDu · 2023-07-01

**Soundness:** 3 good
**Presentation:** 2 fair
**Contribution:** 3 good
**Rating:** 6
**Confidence:** 4

**Summary:**

This paper studies the context shift problem of task representation learning in offline meta-reinforcement learning (OMRL). The proposed method, CSRO, optimizes a combination of FOCAL's objective and an adversarial objective, to maximize task information and minimize behavior policy information in task representations. Experiments in various MuJoCo tasks show that CSRO outperforms baseline methods and learns good representations.

**Strengths:**

1. The problem of context shift in in OMRL is significant. The proposed adversarial method for minimizing the information of behavior policies is novel and makes sense.

2. In experiments, the baselines selected are representative. The test performance of the method is great.

**Weaknesses:**

Some claims in the paper are not very accurate and can be improved:

Line 175: Equation 5 is not equal to the mutual information. It should be explained as an approximation.

Equation 6~8: The meaning of expectation over i and j should be explained.

Line 225: In context collection, taking random actions can also cause context shift to the training distribution. Also, the context collection strategy does not appear to be an original contribution, since CORRO (section 5.6) also uses a random exploration policy to collect context.


**Questions:**

1. Since all the baselines are reimplemented with BRAC, why do the results for CORRO and BOReL are presented with horizontal lines rather than training curves in Figure 3?

2. Is there any difference between FOCAL and the ablation CSRO w/o minMI & Np?

3. Performance of ablation methods is close to CSRO in most experiments according to Figure 4. Does this mean datasets in Half-Cheetah, Humanoid and Hopper cannot reflect the context shift problem?

**Limitations:**

Limitations are not discussed in this paper. I hope the authors address the above issues to improve the paper.

---

> ### Author Rebuttal · Authors · 2023-08-09
>
> We are glad to answer your questions and would appreciate any further response.
>
> > **Q1**: Line 175: Equation 5 is not equal to the mutual information. It should be explained as an approximation.
>
> **A1**:Thank you for your suggestion. This is an approximation, we will modify it later to make it clear.
>
>
>
> > **Q2**: Equation 6~8: The meaning of expectation over i and j should be explained.
>
> **A2**: Here, $z_{i}$ represents  task embedding obtained by $(s_{i}, a_{i}, r_{i}, s_{i}')$ through the context encoder. $E_{j}[\log p(z_{j}|(s_{i},a_{i}))]$ means fixing $i$ and calculating the mean of all $z$. $E_{i}[\log p(z_{i}|(s_{i},a_{i}))\cdots]$ means calculating the mean of each corresponding $z_{i}$ and $(s_{i}, a_{i})$. We will modify it later to make it clearer.
>
>
>
> > **Q3**: Line 225: In context collection, taking random actions can also cause context shift to the training distribution. Also, the context collection strategy does not appear to be an original contribution, since CORRO (section 5.6) also uses a random exploration policy to collect context.
>
> **A3**:There is also some context shift in the random strategy, and in Appendix F we show that this also brings about a slight performance gap. But it produces a smaller distribution shift than the common exploration strategy. The purpose of CORRO is to learn a more robust meta-policy, and then test performance under the context collected by different policies, including random policy. We do it differently, we use warmup data collection with a non-priori random strategy and continuously update the posterior distribution of the context to continue collecting, finding that this can alleviate the context shift.
>
>
>
> > **Q4**: Since all the baselines are reimplemented with BRAC, why do the results for CORRO and BOReL are presented with horizontal lines rather than training curves in Figure 3?
>
> **A4**: This is because CORRO and BOReL train the encoder first, and then train the policy. It is more appropriate to draw a horizontal line. The rest of the methods are to train the encoder and policy at the same time, so it is more appropriate to draw a curve. In this way, it is easier to compare fairly.
>
>
>
> > **Q5**:Is there any difference between FOCAL and the ablation CSRO w/o minMI & Np?
>
> **A5**: FOCAL and CSRO w/o & Np are the same.
>
>
>
> > **Q6**: Performance of ablation methods is close to CSRO in most experiments according to Figure 4. Does this mean datasets in Half-Cheetah, Humanoid and Hopper cannot reflect the context shift problem?
>
> **A6**: In fact, the gap is quite large. From Figure 4, we can see that CSRO has a significant improvement over the CSRO w/o minMI. As for the performance close to CSRO w/o Np, This is because the minMI part has basically solved this problem. In Appendix F, we gave the offline results, as you can see that the performance after using minMI is close to the highest offline performance, and NP has no performance room for improvement.

---

> > ### Comment · Reviewer_DbDu · 2023-08-18
> >
> > Thanks! Your reply addresses most of my concerns. I will keep my initial score.

---

### Official Review · Reviewer_mr2B · 2023-07-03

**Soundness:** 4 excellent
**Presentation:** 4 excellent
**Contribution:** 3 good
**Rating:** 6
**Confidence:** 4

**Summary:**

The paper presents a new method called Context-Shift Robust Offline Meta-Reinforcement Learning (CSRO) to tackle the issue of context shift in offline meta-reinforcement learning. The main contributions of the paper lie in introducing max-min mutual information representation learning during meta-training to lessen the impact of behavior policy, and employing a non-prior context collection strategy during meta-testing to alleviate the impact of the exploration policy. The experimental results demonstrate that CSRO surpasses prior methods in effectively addressing context shift and enhancing performance in demanding domains with reward or dynamic function variations.

**Strengths:**

* The paper is well-written and well-organized.
* This paper addresses an important issue in offline meta RL, specifically the context shift problem that arises due to disparities between training and testing contexts.
* The proposed method introduces a mutual information objective to reduce the reliance of the behavior policy on task representations utilizing FOCAL, and incorporates context-independent random exploration during the initial meta-testing stage.
* Empirical evidence substantiates that the proposed method consistently outperforms other baselines in online test experiments. Additionally, the thorough ablation study validates the effectiveness of the individual components.


**Weaknesses:**

* Regarding the mutual information objective, an additional insight is that $(s,a)$ can be shared across different tasks, while the reward function $r$ plays a vital role in task inference. Equation (8) in the paper focuses the predictions more on the reward rather than solely on $(s,a)$. This insight holds particular significance for point and ant-goal tasks where state-action sharing is more prominent. However, for two Rand-Param tasks, the task can be inferred from the $(s,a)$ pairs, resulting in similar performance between CSRO and CSRO w/o minMI. If this insight holds true, I suggest the authors discuss it in the main paper.
* Another concern is that the non-prior context exploration method directly employs random action exploration, which can be inefficient. Are there other more efficient non-prior context exploration methods that could be utilized for CSRO? The prior exploration method in off-policy meta RL [1] could provide inspiration in this regard.
* To enhance comprehension, it would be beneficial to include a figure in Figure 1 that demonstrates the performance drop of prior works, such as FOCAL, when online exploration is employed to acquire context, as compared to offline context.
* In Figure 5, it appears that CSRO, FOCAL, and CORRO exhibit similar performance. Could you clarify the metric used to compare these three methods? Additionally, to provide a comprehensive understanding, it would be beneficial to include more task representation visualization results in the Appendix, beyond just the HalfCheetah-Vel task.
* Furthermore, it would be advantageous to include a comparison with the recent context correction in offline meta RL [2].

[1] Zhang J, Wang J, Hu H, et al. Metacure: Meta reinforcement learning with empowerment-driven exploration[C]//International Conference on Machine Learning. PMLR, 2021: 12600-12610.

[2] Wang J, Zhang J, Jiang H, et al. Offline Meta Reinforcement Learning with In-Distribution Online Adaptation[J]. arXiv preprint arXiv:2305.19529, 2023.

**Questions:**

* The authors could consider incorporating my insight on the mutual information objective into the main paper for a clearer explanation.
* Are there alternative methods for non-prior context exploration that are more efficient and suitable for CSRO?
* It would be helpful to include a figure in Figure 1 that illustrates the performance drop of prior works like FOCAL when using online exploration to acquire context.
* In Figure 5, where CSRO, FOCAL, and CORRO appear to perform similarly, what metric was used to compare these three methods?
* In addition to the HalfCheetah-Vel task, it would be beneficial to include more task representation visualization results in the Appendix.
* It would be advantageous to include a comparison with the recent context correction work.

**Limitations:**

This paper lacks a discussion on its limitations. One limitation I identified is the inefficient random exploration strategy used during the meta-testing stage with non-prior context.

---

> ### Author Rebuttal · Authors · 2023-08-09
>
> Thanks a lot for your advice on further improving this paper. We would like to discuss them one by one.
>
> > **Q1**: The authors could consider incorporating my insight on the mutual information objective into the main paper for a clearer explanation.
>
> **A1**: Thank you very much for your suggestion. We will modify the mutual information part in the future so that it can be explained better. First of all, ideally, the agent should pay more attention to the reward function of the environment where the reward changes, and the dynamic function of the environment where the dynamic changes, and only infer the task from there. However, our behavior policy and task are highly correlated. Both the environment of reward change and the environment of dynamic change can infer the task from $(s,a)$, but the policy and task during the test are not related, which will cause context shift problems, so we need Mutual information processing.
>
> Regarding the Rand-Param tasks you mentioned, because we use medium data, there is some interference with task inference. In addition, if the agent samples the same $(s,a)$ as the training environment when exploring the test environment, this will also cause interference, so it is still necessary to minimize the mutual information. In the ablation experiments in Figure 4, minMI improves the performance of both environments.
>
>
>
> > **Q2**: Are there alternative methods for non-prior context exploration that are more efficient and suitable for CSRO?
>
> **A2**:We add the MetaCURE method to CSRO and use the offline dataset to train the exploration policy. The experimental results are as follows:
>
> | Env           | Point-Robot | Half-Cheetah-Vel |
> | ------------- | ----------- | ---------------- |
> | CSRO          | -6.4        | -48.4            |
> | CSRO+MetaCURE | -13.2       | -87.9            |
>
> We can see that experimenting with MetaCURE does not work well. Because in offline settings, there is no way to interact with the environment, and using offline datasets to train the exploration policy, the exploration policy is limited to the vicinity of the datasets, and the conservatism of offline is in conflict with exploration, so by training an exploration policy to solve this problem in offline settings is difficult. We will discuss in the paper why the way of training exploration strategies in online meta RL is difficult to use in offline meta RL, and cite this article.
>
>
>
> > **Q3**: It would be helpful to include a figure in Figure 1 that illustrates the performance drop of prior works like FOCAL when using online exploration to acquire context.
>
> **A3**: Thank you for your suggestion. The two test results of FOCAL and OffPearl are given below, and the obvious performance degradation can be seen.  We will add it to the paper later to enhance comprehension.
>
> | Env      | Point-Robot |        | Half-Cheetah-Vel |        |
> | -------- | ----------- | ------ | ---------------- | ------ |
> |          | offline     | online | offline          | online |
> | FOCAL    | -4.4        | -14.9  | -45.7            | -69.5  |
> | OffPearl | -5.1        | -17.8  | -123.0           | -162.8 |
>
>
>
>
> > **Q4**: In Figure 5, where CSRO, FOCAL, and CORRO appear to perform similarly, what metric was used to compare these three methods?
>
> **A4**: The metric we use to compare several methods is whether the task embeddings of the same task can be clustered together and whether different tasks can be distinguished. In Figure 5, although FOCAL and CORRO cluster the same tasks together, the degree of differentiation of different tasks is not as good as CSRO. Similar colors in Figure 5 represent similar tasks. We can see that similar tasks of FOCAL and CORRO are connected together.
>
>
>
> > **Q5**: In addition to the HalfCheetah-Vel task, it would be beneficial to include more task representation visualization results in the Appendix.
>
> **A5**: Thank you for your suggestion. We added the t-sne visualization of CSRO, FOCAL, and CORRO in the Point-Robot in the pdf of the global response. We can see that FOCAL is worse. In CORRO, similar tasks are closer, and many points of the same task are far apart. We will add more visualizations of the environment in the appendix.
>
>
>
> > **Q6**: It would be advantageous to include a comparison with the recent context correction work
>
> **A6**: We reproduced the results of IDAQ on our offline datasets, and we can see that CSRO and IDAQ are comparable. We will add this experiment later, citing this article.
>
> | Env  | Point-Robot | Half-Cheetah-vel | Walker-Rand-Params |
> | ---- | ----------- | ---------------- | ------------------ |
> | CSRO | -6.4        | -48.4            | 344.2              |
> | IDAQ | -5.2        | -60.9            | 297.0              |

---

> > ### Comment · Reviewer_mr2B · 2023-08-11
> >
> > While most of my concerns have been addressed, I still have reservations regarding the inefficient random exploration strategy employed during the meta-testing stage. Although the authors have acknowledged the challenges of training exploration strategies in offline meta RL compared to online meta RL, this concern remains. Therefore, I would keep my current score.

---

### Official Review · Reviewer_4mWP · 2023-07-05

**Soundness:** 3 good
**Presentation:** 2 fair
**Contribution:** 2 fair
**Rating:** 5
**Confidence:** 4

**Summary:**

This manuscript proposes Context Shift Reduction (CSRO) for the offline meta reinforcement learning problem. It aims at solving the context shift problem with only offline datasets, and demonstrates superior empirical performance.

**Strengths:**

The paper is easy to understand and the experiments look reasonable.

**Weaknesses:**

One major weakness the reviewer identifies is limited discussion of the disadvantages of prior methods / or major novelties of the proposed method. For example, there are several potential improvements the authors can take:
1. Provide theoretical justifications for the proposed method, such as under what condition the method can outperforms other methods
2. Provide intuition on why prior methods do not solve the context shift problem well enough
3. What are the main benefits/novelties of the CSRO method in solving context shift problem

Overall the reviewer thinks the empirical results look solid and promising, and the reviewer would like to adjust the rating if the author can adjust the writing to better present the proposed method. Minor presentation issues:
1. There is a missing space in line 78, between “[6,28]” and “methods”.



**Questions:**

1. Why is BRAC chosen as the offline backbone algorithm instead of CQL [1] or IQL [2]?
2. If the reviewer understands correctly, in line 125 the context is defined as a subset from the offline dataset $\{(s_j,a_j,r_j,s_j’)\}_{j=1}^n$. Is there any specific the context need to defined in this way rather than defined in a general context space?

[1] Kumar, Aviral, et al. "Conservative q-learning for offline reinforcement learning." Advances in Neural Information Processing Systems 33 (2020): 1179-1191.
[2] Kostrikov, Ilya, Ashvin Nair, and Sergey Levine. "Offline reinforcement learning with implicit q-learning." arXiv preprint arXiv:2110.06169 (2021).

**Limitations:**

See weaknesses and questions.

---

> ### Author Rebuttal · Authors · 2023-08-09
>
> Thank you for providing your comprehensive review. We greatly appreciate your insights and are glad to address each of your concerns in a detailed manner.
>
> > **Q1**: Provide theoretical justifications for the proposed method, such as under what condition the method can outperforms other methods
>
> **A1**: Denote context $c=\\{(s,a,r,s')\\}$ as an experience collected by explore policy $\\pi_{e}$, test task $M_{i}=(S,A,P_{i},\\rho, R_{i})\\sim p(M)$. The expected return of test task $M_{i}$ which is evaluated by a learned meta-policy $\\pi_{\\theta}(a|s,z)$ is
>  $J_{M_{i}}(\\pi_{\\theta},\\pi_{e})=E_{s_{0}\\sim\\rho(s_{0}),z\\sim q_{\\phi}(\\cdot|c),a_{t}\\sim\\pi(\\cdot|s_{t},z), r_{t}\\sim R_{i}(\\cdot|s_{t},a_{t}),s_{t}'\\sim P(\\cdot|s_{t},a_{t})}[\\sum_{t=0}^{H-1}r_{t}]$.
>  The meta-policy gets the highest expected return when explore policy $\\pi_{e}=\\mu_{i}$. However, for any exploration policy $\\pi_{e}$, we hope $J_{M_{i}}(\\pi_{\\theta},\\pi_{e})=J_{M_{i}}(\\pi_{\\theta},\\mu_{i})$, if and only if for any $\\pi_{e}$, $q_{\\phi}(z|c)$ is the same.
>
> Since $c$ is collected by $M_{i}$ and $\\pi_{e}$, $q_{\\phi}(z|c)$ can be written as $q_{\\phi}(z|M_{i},\\pi_{e})$, i.e. for any $\\pi_{e}$, $q_{\\phi}(z|M_{i},\\pi_{e})$ is the same. So $z$ and $\\pi_{e}$ are independent, that is, the mutual information $I(z;\\pi_{e})=0$.
>
> Therefore, we minimize the mutual information between the policy and $z$ to alleviate the context shift problem. The mutual information upper bound on minimizing mutual information has been proved in the paper "Efficient off-policy meta-reinforcement learning via probabilistic context variables".
>
> In addition to the offline datasets, other methods also require additional information related to the environment, which does not solve the problem well. We focus on the case of using only offline datasets and alleviate this problem by training a more essential encoder and a more appropriate collection context strategy, so our solution is better than the mentioned method.
>
>
>
> > **Q2**: Provide intuition on why prior methods do not solve the context shift problem well enough
>
> **A2**: As for why the prior methods did not solve this problem, we briefly described it in the introduction.
>
> Specifically, FOCAL does not consider this problem and directly uses the pre-collected context in the test environment to test evaluation. BOReL thinks that the problem comes from MDP ambiguity, by assuming that the reward function of all tasks is known and using the reward function to relabel different task data to solve it. SMAC thinks that the problem comes from the policy of collecting context during training is different from that of testing. It is solved by destroying the complete offline setting and performing some online training. The latter two methods require additional information beyond the use of offline datasets alone, and the solutions are not ideal, we hope to solve this problem with only offline datasets. We will revise the article later with a more detailed description.
>
>
>
> > **Q3**: What are the main benefits/novelties of the CSRO method in solving context shift problem
>
> **A3**: Unlike prior methods that used information beyond offline datasets, our proposed method uses only offline datasets. We propose that the cause of the problem is the influence of policy information on task embeddings, and it is not enough to maximize the mutual information between the environment and task embedding, and it is also necessary to minimize the mutual information between the policy and task embedding. We empirically find that a non-a prior context collection strategy consisting of a small number of random explorations can alleviate this problem.
>
>
>
> > **Q4**: There is a missing space in line 78, between "[6,28]" and "methods".
>
> **A4**: Thanks for your advice. We will fix it later.
>
>
>
> > **Q5**: Why is BRAC chosen as the offline backbone algorithm instead of CQL or IQL?
>
> **A5**: Because the baseline method FOCAL uses BRAC, for a fair comparison, we also use BRAC.
>
>
>
> > **Q6**: If the reviewer understands correctly, in line 125 the context is defined as a subset from the offline dataset $(s_{j},a_{j},r_{j},s_{j}')^{n}_{j=1}$. Is there any specific the context need to defined in this way rather than defined in a general context space?
>
> **A6**: Your understanding is correct, context is defined as a subset of offline datasets. This is because we only have collected offline datasets to train the encoder, so the context is defined as a subset of the offline datasets. The prior work is also defined in this way, such as "Robust Task Representations for Offline Meta-Reinforcement Learning via Contrastive Learning".

---

> > ### Comment · Reviewer_4mWP · 2023-08-12
> > **Response to the Rebuttals**
> >
> > Dear Authors,
> >
> > Thank you very much for your responses. The reviewer has no more other questions, and since all of my questions/concerns are adequately addressed, I have adjusted my rating accordingly.
> >
> > Best,
> > Reviewer 4mWP

---

### Author Rebuttal · Authors · 2023-08-09

Dear reviewer, PDF has our supplementary experimental diagram.

---

### Decision · Program_Chairs · 2023-09-21

**Decision:**

Accept (poster)

**Comment:**

This paper studies an important problem in offline meta RL, and proposes a novel and very sensible approach. All the reviewers agreed to accept the paper.

I have a comment to the authors in respect to the comparison with BOREL. First, BOREL (and similarly VariBAD) are trained to be Bayes-optimal, while the proposed method is based on Thompson sampling (and also includes random exploration at test, which is even less optimal). This means that potentially, if context shift wasn’t a problem, BOREL could reach  much higher scores at test time (see the BOREL paper for a discussion and comparison to PEARL). This should be more clearly mentioned in the related work, as from the current writing it appears that BOREL and CSRO solve the same problem, which is not correct. Another difference is that CSRO requires knowing the task labels in the training data, while BOREL does not (yet BOREL requires either the reward functions, or to collect data in a very specific way). The two points above make the comparison with BOREL less meaningful, and the authors should clarify it in the text.